# Yeast Models of Amyotrophic Lateral Sclerosis Type 8 Mimic Phenotypes Seen in Mammalian Cells Expressing Mutant VAPB^P56S^

**DOI:** 10.3390/biom13071147

**Published:** 2023-07-19

**Authors:** AnnaMari L. Stump, Daniel J. Rioux, Richard Albright, Guiliano L. Melki, Derek C. Prosser

**Affiliations:** 1Department of Biology, Virginia Commonwealth University, Richmond, VA 23284, USA; 2VCU Life Sciences, Virginia Commonwealth University, Richmond, VA 23284, USA

**Keywords:** amyotrophic lateral sclerosis, ALS, ALS8, endoplasmic reticulum, membrane contact site, neurodegeneration, *Saccharomyces cerevisiae*, VAPB, yeast

## Abstract

Amyotrophic lateral sclerosis (ALS) is a complex neurodegenerative disease that results in the loss of motor neurons and can occur sporadically or due to genetic mutations. Among the 30 genes linked to familial ALS, a P56S mutation in *VAPB*, an ER-resident protein that functions at membrane contact sites, causes ALS type 8. Mammalian cells expressing VAPB^P56S^ have distinctive phenotypes, including ER collapse, protein and/or membrane-containing inclusions, and sensitivity to ER stress. *VAPB* is conserved through evolution and has two homologs in budding yeast, *SCS2* and *SCS22*. Previously, a humanized version of *SCS2* bearing disease-linked mutations was described, and it caused Scs2-containing inclusions when overexpressed in yeast. Here, we describe a yeast model for ALS8 in which the two *SCS* genes are deleted and replaced with a single chromosomal copy of either wild-type or mutant yeast *SCS2* or human *VAPB* expressed from the *SCS2* promoter. These cells display ER collapse, the formation of inclusion-like structures, and sensitivity to tunicamycin, an ER stress-inducing drug. Based on the phenotypic similarity to mammalian cells expressing VAPB^P56S^, we propose that these models can be used to study the molecular basis of cell death or dysfunction in ALS8. Moreover, other conserved ALS-linked genes may create opportunities for the generation of yeast models of disease.

## 1. Introduction

Neurodegenerative disorders comprise the set of diseases in humans that result in the loss of central and/or peripheral nervous system function due to neuronal death. Oftentimes, neurodegenerative diseases selectively affect neurons in specific brain regions, leading to symptoms characteristic of Parkinson’s disease (substantia nigra), Huntington’s disease (striatum), Alzheimer’s disease (hippocampus) and amyotrophic lateral sclerosis (motor cortex), among others. Both genetic and environmental factors can contribute to neurodegeneration, leading to numerous mechanisms for neuronal death that are not fully understood.

Amyotrophic lateral sclerosis (ALS) is a particularly complex neurodegenerative disease, in which upwards of 30 genes have been linked to motor neuron death, leading to muscle weakness and loss of voluntary movement [1,2,3]. Mutations in these genes lead to familial ALS, which constitutes only ~10% of the total cases; the remaining ~90% are idiopathic in nature and referred to as sporadic ALS. Genes linked to ALS have been implicated in a variety of functions, including RNA processing, transport and stabilization [4,5,6], mitochondrial function [7,8,9], protein and membrane traffic, nuclear function [10,11], ER stress and protein quality control [1,12,13,14,15,16]. Thus, it appears that motor neuron death in ALS may occur for numerous reasons. Interestingly, cells isolated from sporadic ALS patients often show phenotypes that are similar to those seen in familial cases, suggesting the possibility of mechanistic similarities for cellular dysfunction and neurodegeneration in sporadic and familial ALS [17]. Although familial ALS comprises only a small proportion of the total cases, these phenotypic similarities suggest that mechanistic studies of neurodegeneration based on the genetic causes of ALS can offer broader insights into motor neuron loss. Advances in therapeutic development based on familial ALS might improve outcomes and quality of life for larger groups of patients. Developing genetically tractable models to study cellular dysfunction, toxicity and the molecular mechanisms of motor neuron death is thus a high priority in the ALS field.

Our interest in understanding the cellular basis of neurodegeneration arises from prior studies of vesicular-associated membrane protein (VAMP)-associated protein B VAPB and its closely related isoform, VAPA [18]. VAPs were originally identified as proteins with predicted similarity to the VAMP family of v-SNAREs involved in vesicle fusion, bearing a C-terminal transmembrane domain and a cytoplasmic coiled-coil region typical of SNAREs [19,20]. They additionally possess an N-terminal major sperm protein (MSP) domain that binds to proteins with an FFAT (two phenylalanines in an acidic tract) motif, including oxysterol-binding protein (OSBP), OSBP-related proteins (ORPs) and ceramide transfer protein (CERT), involved in nonvesicular lipid shuttling [21,22,23]. Both VAPA and VAPB localize to the endoplasmic reticulum (ER) and are major proteins at membrane contact sites between the ER and numerous organelles, such as the plasma membrane (PM), endosomes, Golgi complex and mitochondria [24,25,26,27].

A P56S mutation within the MSP domain of VAPB was originally identified as the causative mutation in a Brazilian family with autosomal dominant ALS type 8 (ALS8) [28]. VAPB^P56S^ is predicted to alter the folding and stability of the MSP domain, and biochemical analyses revealed that the mutant protein is unstable and aggregate-prone [28,29,30]. Accordingly, VAPB^P56S^ causes ER collapse and the formation of inclusions that may or may not contain other ER-resident proteins depending on the study and/or the cell line used [18,31]. Evidence suggests that mutant VAPB^P56S^ acts in a dominant-negative manner, perhaps by sequestering wild-type VAPA and VAPB into inclusions [13,31,32]. In addition to altering ER morphology, the expression of VAPB^P56S^ in mammalian cells impairs the anterograde transport of transmembrane proteins from the ER to the Golgi, likely by impeding their lateral diffusion and incorporation into ER exit sites [18]. A deficit in cargo exit could result in ER stress and may also impact motor neuron function or survival by reducing the axonal transport of proteins that act at synapses, including cell surface receptors that play trophic roles. Alternatively, VAPB may be processed by proteolysis, and a secreted form has been suggested to play trophic roles by binding to Eph receptors [33].

VAPB^P56S^ may also lead to toxic effects by modifying the unfolded protein response (UPR), which is activated during ER stress [12,13,14]. When misfolded proteins accumulate in the ER, UPR activation pauses translation and increases the activity of chaperones to assist with protein folding. The UPR consists of three signaling pathways that are activated in the presence of unfolded proteins. First, protein kinase R (PKR)-like ER kinase (PERK) phosphorylates eIF2α to block translation and increase the expression of activating transcription factor 4 (ATF4) [34,35]. Second, inositol-requiring enzyme 1 (IRE1) dimerizes, leading to the splicing of a 26-nucleotide intron from the mRNA of the stress-responsive transcription factor X-box binding protein 1 (XBP1), yielding a spliced form (XBP1s) [36]. Finally, ATF6 is a transmembrane protein that translocates from the ER to the Golgi during ER stress, where it undergoes proteolytic cleavage to release a cytoplasmic N-terminal region that translocates to the nucleus [37]. ATF4, XBP1s and the soluble N-terminal fragment of ATF6 then act as transcription factors that upregulate UPR target genes. Previous studies showed that the overexpression of wild-type or P56S mutant VAPB attenuated the ATF6 branch of the UPR [12]. In contrast, the overexpression of VAPB^WT^ activates the IRE1/XBP1 branch of the UPR, while VAPB^P56S^ does not [13,14]. Taken together, ER collapse, protein misfolding/aggregation and the aberrant regulation of ER stress responses may each contribute to motor neuron death in ALS8, although the precise mechanisms governing this process are not fully understood.

The budding yeast *Saccharomyces cerevisiae* is a powerful model organism for the study of many conserved aspects of cellular function. Yeast have two homologs of mammalian *VAP* genes, known as *SCS2* and *SCS22*, that also reside in the ER and play prominent roles in regulating membrane contact sites and ER stress responses [24,25]. Initial studies suggested that the Scs2^P51S^ mutant, which is analogous to VAPB^P56S^, remained functional in yeast [22]. However, additional humanizing mutations (Scs2^P44S, P51S^ or Scs2^P51S, P58S^) caused the protein to accumulate in perinuclear inclusion-like structures, similar to those seen in mammalian cells expressing VAPB^P56S^, and resulted in loss-of-function phenotypes including inositol auxotrophy when expressed in *scs2*∆ cells [38]. In this study, we generate yeast models to study ALS8 by deleting *SCS2* and *SCS22* to ablate their function and re-introducing a single chromosomal copy of wild-type or ALS8-linked alleles of either yeast *SCS2* or human *VAPB*. While *scs2*∆ *scs22*∆ cells show ER collapse and stress sensitivity, similar to mammalian cells expressing VAPB^P56S^, we find that wild-type Scs2 or VAPB ameliorates these phenotypes, whereas mutant Scs2^P51S, P58S^ or VAPB^P56S^ is less effective or ineffective. Based on our findings, we propose that these yeast models may yield insights into the cellular basis of toxicity arising from ALS8-linked mutations. Moreover, the genetic tractability of yeast can provide opportunities to identify conserved genes that suppress disease-related phenotypes, which may thus represent therapeutic targets.

## 2. Materials and Methods

### 2.1. Yeast Strains, Growth Conditions, and Reagents

Yeast strains used in this study are listed in Table 1. All strains were derived from W303 (*MATa leu2-3,112 trp1-1 can1-100 ura3-1 his3-11,15*) as the wild-type (WT) parental strain. Cells were cultured in yeast extract/peptone/dextrose (YPD) medium or on defined yeast nitrogen base (YNB) medium containing ammonium sulfate and 2% (*w*/*v*) glucose. YNB medium was supplemented with histidine, leucine, tryptophan, uracil, lysine, methionine and adenine (YNB complete), while individual amino acids or bases were omitted from the medium for the selection of transformed plasmids. Unless otherwise stated, cells were grown at 30 °C on sterile YPD or YNB plates containing 2% (*w*/*v*) agar in a gravity convection incubator (VWR, Radnor, USA), or in liquid medium using an orbital shaking incubator at 250 rpm (MaxQ™ 8000, Thermo Fisher Scientific, Waltham, MA, USA).

Gene deletions and C-terminal tagging with fluorescent proteins at endogenous gene loci were performed using PCR-based integration methods, as described previously [39,40]. Transformation of plasmids or PCR-based integrating cassettes was performed using the lithium acetate method [41], with the selection of transformants on appropriate YNB plates or on YPD plates supplemented with G418, nourseothricin or hygromycin B.

Unless otherwise stated, all reagents were purchased from Fisher Scientific or Sigma Aldrich (St. Louis, MO, USA).

### 2.2. Plasmids

Plasmids used in this study are listed in Table 2. The *SCS2* coding sequence, along with 500 bp of upstream and downstream sequences containing the *P_SCS2_* promoter and *T_SCS2_* terminator, respectively, was amplified by PCR using genomic DNA isolated from SEY6210 (*MATα leu2-3,112 ura3-52 his3-*∆*200 trp1-*∆*901 lys2-801 suc2-*∆*9*) as a template and cloned into pRS413 (*CEN HIS3*) by homologous recombination in WT yeast. Primers were designed to contain sequences homologous to the multiple cloning site of pRS413 at the ends of the *SCS2* PCR product. s*cs^2P51S, P58S^* was generated by site-directed mutagenesis and cloned into pRS413 as described for *SCS2^WT^*. To generate integrating plasmids, inserts containing *SCS2* or *scs2^P51S, P58S^* along with their promoter and terminator sequences were excised by digestion with NotI and SalI (New England Biolabs, Ipswich, MA, USA) and subcloned into the NotI and SalI sites of pRS405 (*LEU2*). To facilitate downstream detection of the *scs2^P51S, P58S^* allele, we included silent mutations that created a new PvuI site proximal to the P51 residue and a new NheI site proximal to the P58 residue (Appendix A).

Human *VAPB* and *VAPB^P56S^* cloned into pFLAG-CMV2 were kindly provided by Dr. Johnny K. Ngsee (University of Ottawa) [18]. These were used as templates for PCR amplification and subcloning into pRS416 (*CEN URA3*). Briefly, three fragments consisting of (1) 500 bp of yeast genomic DNA immediately upstream of the *SCS2* start codon; (2) the coding sequence of human *VAPB* or *VAPB^P56S^*; and (3) 500 bp of yeast genomic DNA immediately downstream of the *SCS2* stop codon were amplified by PCR using primers that created an overlap between each adjacent fragment or between the *SCS2* untranslated regions and the multiple cloning site of pRS416 [42]. The three fragments were assembled and cloned into pRS416 by homologous recombination in WT yeast. To generate integrating plasmids, inserts containing WT or mutant *P_SCS2_-VAPB-T_SCS2_* were subcloned into the NotI and SalI sites of pRS405, as described above for *SCS2* [42]. The Scs2 and VAPB plasmids were confirmed by Sanger sequencing, and the coding sequences used are listed in Appendix A.

Genomic integration of empty pRS405 or pRS405 plasmids containing *SCS2* or *VAPB* sequences into the *LEU2* 5′ untranslated region was performed by linearizing each plasmid with NarI (New England Biolabs), transforming them into yeast strains as described above and selecting them on YNB-Leu medium.

### 2.3. Plate-Based Growth Assays

To assess the sensitivity of the yeast strains to ER stress, plate-based growth assays were performed in which equivalent volumes of serially diluted cells grown to the mid-logarithmic phase were plated on YNB complete medium with increasing concentrations of tunicamycin (Abcam, Cambridge, UK). Briefly, cells were inoculated in YNB complete medium and grown overnight at 30 °C in a shaking incubator. Culture densities were measured using a Genesys 10 s Vis spectrophotometer (Thermo Fisher Scientific, Waltham, MA, USA); cells were diluted to a density of 0.4 OD_600_/mL in YNB complete medium, and growth was monitored until cells reached the mid-logarithmic phase (0.6–0.8 OD_600_/mL). Cells were then diluted to a starting volume of 1.0 OD_600_/mL and subjected to a five-fold serial dilution series (1:5, 1:25 and 1:125). Equal volumes of each dilution were transferred into a 32-well pinning replicator plate (Dan-Kar Corp., Woburn, MA, USA), and a matched pronging device was used to spot cells onto triplicate YNB complete plates with 0, 0.1875, 0.25, 0.375 or 0.5 µg/mL tunicamycin (Abcam product ab120296, MW = 844.95 g/mol) diluted from a 5 mg/mL stock dissolved in DMSO. Plates were grown at 30 °C for 4–6 days prior to imaging with a FluorChem M gel documentation system (Bio-Techne, Minneapolis, MN, USA).

### 2.4. Kinetic Growth Assays

To assess the kinetics of growth in WT and mutant yeast strains, we established a liquid culture-based assay for cells grown in a 48-well dish. Cells were inoculated in YNB complete medium and grown for approximately 24 h in a 30 °C shaking incubator. Culture densities were measured using a 10 s Vis spectrophotometer, and cells were diluted to a starting density of 1.0 OD_600_/mL. For each strain tested, a series of seven wells in a 48-well dish were prepared with 500 µL of YNB complete medium supplemented with 0, 0.125, 0.25, 0.375, 0.5, 0.75 and 1 µg/mL tunicamycin diluted from a 5 mg/mL stock dissolved in DMSO. To each well, 30 µL of diluted cells were added, and the plate was transferred to a SpectraMax i3x multi-mode plate reader controlled by the SoftMax Pro v7.1.0 software (Molecular Devices, San Jose, CA, USA). Each plate additionally contained a blank well with 500 µL of YNB complete medium, but without cells. Growth kinetics were monitored by reading OD_600_ values every 15 min for 16 h, with continuous shaking between reads.

Raw data from each experiment were exported to Microsoft Excel and processed as follows. First, absorbance from the blank well was subtracted from each experimental well. Subsequently, absorbance in each well was normalized to the same starting density of 0.056604 OD_600_/mL based on the dilution of 30 µL of cells at 1.0 OD_600_/mL into a final volume of 530 µL, and this normalization was used to create a correction factor that was applied to all time points for the well. Growth curves were then plotted using Prism version 7.0d (GraphPad, San Diego, CA, USA).

From these processed results, we estimated the IC_50_ of tunicamycin in WT and mutant strains by normalizing the relative growth at 12 h for strains in each drug concentration to the same cells grown in the absence of tunicamycin, since not all cells had reached the stationary phase at that time point. Using these data, we plotted a dose–response curve fitted on the mean of five independent trials based on the log[inhibitor] versus normalized response (variable slope) model in Prism 7.0d, and we determined the IC_50_ from this model. We also used the normalized data from the same trials to compare growth in 0.5 µg/mL tunicamycin at 12 h relative to no drug between WT and mutant strains.

### 2.5. Fluorescence Microscopy and Image Processing

All fluorescence microscopy experiments were performed using a DMi8 inverted fluorescence microscope (Leica Microsystems, Wetzlar, Germany) equipped with a 100×, 1.47 numerical aperture (NA) Plan-Apochromat oil immersion lens, a Flash 4.0 v3 sCMOS camera (Hamamatsu, Shizuoka, Japan), an LED3 fluorescence illumination system, 488 nm and 561 nm lasers, a W-View Gemini image splitting optical device (Hamamatsu), compatible filter sets for fluorescence and DIC imaging and the LAS X v3.7.6.25997 software (Leica). Image acquisition parameters (illumination intensity, exposure time and camera binning) were consistently applied within each trial of an experiment to allow comparison between strains.

Images were processed following acquisition by importing them into Fiji/ImageJ2 v2.9.0/1.53t. Linear adjustments were made to minimum and maximum intensity levels for each 16-bit image within an experiment prior to conversion into 8-bit format for figure preparation.

### 2.6. Quantification of ER Inclusion Size

To quantify the size of the ER inclusion-like structures, images were acquired as described above, where random fields of cells were selected using only the DIC mode of the microscope, to avoid selection bias for ER morphology based on the localization of Sec61-GFP. The 16-bit images were imported into Fiji/ImageJ2, and the lasso tool was used to select and measure the area of the largest inclusion-like structure in each cell. Inclusion-like structures were defined as the largest structure containing Sec61-GFP that was distinct from the nuclear and cortical ER. A minimum of 100 cells per independent trial were measured for each strain. For each image used in the analysis, inclusion-like structures were measured for all cells except for (1) those that were touching the edge of the image in such a way that the entire cell could not be seen; (2) cells that were out of focus; and (3) cells that were dead/dying as assessed by DIC [44].

### 2.7. Comparison of Human ALS-Linked Proteins with Putative Yeast Homologs

Human ALS-linked genes and disease-causing alleles were determined based on annotation in Online Mendelian Inheritance in Man (OMIM). Protein sequences for human genes were obtained from the National Center for Biotechnology Information (NCBI) protein database, with accession numbers listed in Table 3. Yeast homologs were identified using the Alliance of Genome Resources v5.4.0 or the *Saccharomyces Genome Database*, and all yeast protein sequences were obtained from the *Saccharomyces Genome Database* (www.yeastgenome.org; accessed on 9 May 2023) based on the S288C reference strain. Sequence alignments and percentage identity were assessed using Clustal Omega.

### 2.8. Statistical Analysis and Generation of SuperPlots

To display the quantitative analysis of the ER inclusion size, we generated SuperPlots, as described previously [109]. Briefly, SuperPlots identify individual measurements within each independent trial by color and/or shape coding as the background of a scatter plot. Above this information are overlaid trial-level summary data with the same color/shape coding. For our plots, we coded independent trials with blue circles, yellow squares or gray triangles. Trial-level summary points represent the median of each trial because samples showed a tailed distribution. Bars correspond to the mean ± 95% confidence interval computed based on the trial medians.

For the quantification of the ER inclusion-like structure size, statistical analysis was performed using a nested one-way ANOVA followed by Tukey’s multiple comparison test in Prism 9. For comparison of the 12 h relative growth levels, statistical analyses were performed using repeated-measures one-way ANOVA followed by Dunnett’s multiple comparison test relative to WT or *scs2*∆ *scs22*∆ cells as controls.

## 3. Results

### 3.1. Loss of SCS2 and SCS22 Causes ER Collapse and Sensitivity to ER Stress

To establish our yeast model of ALS8, we first examined phenotypes associated with the single or pair-wise deletion of the VAPB homologs *SCS2* and *SCS22*. Both Scs2 and Scs22 have known functions as tethering proteins at ER–PM junctions, along with Ist2, Tcb1, Tcb2 and Tcb3 [24]. Previous studies have shown that *scs2*∆ and *scs2*∆ *scs22*∆ cells show a reduction in cortical ER [24,25]; however, to our knowledge the effect of *scs22*∆ alone on ER morphology has not been reported. Thus, we generated *scs2*∆, *scs22*∆ and double *scs2*∆ *scs22*∆ mutant strains expressing Sec61-GFP from its chromosomal locus as an ER-resident component of the translocon complex. In WT cells, Sec61-GFP prominently localized to the nuclear ER (nER) and to the cortical ER (cER), with occasional small gaps in cortical coverage (Figure 1A). In contrast, both *scs2*∆ and *scs2*∆ *scs22*∆ cells displayed regions of the cell cortex with large gaps lacking cER attachment (Figure 1A, arrows), although the phenotype appeared to be more pronounced in *scs2*∆ *scs22*∆ cells since cER could still be readily observed in *scs2*∆ cells. Deletion of *SCS22* alone did not alter the ER morphology compared to WT, suggesting that the loss of Scs22 alone does not have a pronounced effect on the formation of ER–PM tethers.

When we examined Sec61-GFP localization in *SCS* loss-of-function cells, we noticed the appearance of brighter inclusion-like structures, most frequently in the perinuclear region, which are typical of a collapsed ER (Figure 1A, arrowheads) [24]. To quantify the extent of ER collapse and/or the formation of inclusion-like structures, we measured the area of the largest inclusion-like structure in each cell that was distinct from the nER and cER. Using this approach, we found that the inclusion-like structures were similar in size for WT, *scs2*∆ and *scs22*∆ cells; although *scs2*∆ cells showed a trend toward slightly larger inclusions, this was not a statistically significant difference (*p* = 0.1283; Figure 1B). In contrast, *scs2*∆ *scs22*∆ double mutant cells had significantly larger inclusion-like structures than WT cells, consistent with a collapsed ER morphology. Thus, Scs2 and Scs22 appear to play overlapping roles in maintaining cER in yeast, and the loss of both proteins simultaneously leads to the pronounced loss of cER.

Mammalian cells expressing VAPB^P56S^ show aberrant regulation of the UPR, as seen by impaired ATF6- and IRE1/XPB1-dependent transcriptional activity, which is required for the expression of downstream UPR genes [13,14,110]. The reduced ability of VAPB^P56S^-expressing cells to activate the UPR may increase the sensitivity to ER stress due to an inability to properly clear misfolded proteins, which in turn could at least partially account for the neurodegeneration in ALS8. Consistent with this idea, ∆tether yeast cells lacking proteins required for ER–PM tethering (including Scs2 and Scs22) are sensitive to the ER stress-inducing drug tunicamycin, which inhibits the Alg7 protein involved in the first step of N-linked glycosylation [24,111]. Moreover, ∆tether cells show pronounced growth defects when combined with deletions of *IRE1* or *HAC1*, the yeast equivalents of mammalian IRE1 and XBP1 [24]. Based on these results, we assessed the ER stress sensitivity of WT, *scs2*∆, *scs22*∆ and *scs2*∆ *scs22*∆ cells by growing equivalent volumes of serially diluted cells on plates with increasing concentrations of tunicamycin (Figure 1C). Both WT and *scs22*∆ cells began to display sensitivity to tunicamycin at a concentration between 0.375 and 0.5 µg/mL, but grew robustly at lower concentrations or in the absence of the drug. In contrast, *scs2*∆ and *scs2*∆ *scs22*∆ cells were mildly sensitive to tunicamycin, even at the lowest concentration tested (0.1875 µg/mL), where the spots of cells were slightly less dense than WT or *scs22*∆ cells. The increased sensitivity to tunicamycin became more pronounced at a higher drug concentration, suggesting that the loss of *SCS2* was sufficient to render cells sensitive to ER stress. Serially diluted spots of *scs2*∆ *scs22*∆ showed slightly sparser growth than *scs2*∆ cells at 0.375–0.5 µg/mL tunicamycin; thus, the complete loss of *SCS* function in the double mutant may result in a slightly more severe phenotype than *scs2*∆ alone. Notably, we found that the expression of Sec61-GFP increased tunicamycin sensitivity compared to untagged cells, even in the WT background. As a result, we used untagged cells for all growth assays in this study.

Taken together, these results agree with previous studies showing that Scs2 and Scs22 play roles in ER morphology and in the regulation of ER stress responses [24,25]. Our observation that the phenotypes in *scs2*∆ *scs22*∆ double mutant cells were more pronounced than in *scs2*∆ or *scs22*∆ single mutants supports the overlapping nature of Scs2 and Scs22 function. Since *scs2*∆ alone rendered cells sensitive to tunicamycin and caused gaps in cER, Scs2 may represent the major isoform in yeast, with a smaller contribution from, or lower expression of, Scs22. Indeed, numerous comparative expression studies have shown that Scs2 is considerably more abundant than Scs22, and the half-life of Scs2 is also estimated to be greater than that of Scs22 [112,113,114,115,116,117,118]. These data prompted us to use *scs2*∆ *scs22*∆ as the basis of our yeast models for ALS8 in order to avoid potential confounding impacts of *SCS22* expression.

### 3.2. Expression of SCS2, but Not scs2^P51S, P58S^, Corrects ER Morphology and Stress Sensitivity Defects in scs2∆ scs22∆ Cells

In the first version of our yeast ALS8 model, we wanted to restore the expression of *SCS2* in *scs2*∆ *scs22*∆ cells using either a wild-type allele or a humanized allele bearing ALS8-linked mutations. While VAPB^P56S^ causes ER collapse, inclusion formation and aberrant ER stress responses in mammalian cells, initial studies showed that the analogous P56S mutation in VAPA has a reduced or no effect on ER morphology [18,31]. Similarly, the analogous yeast Scs2^P51S^ mutant retains the ability to bind peptides with an FFAT motif [22], suggesting that this mutation is unique in leading to disease phenotypes when present in *VAPB*, but not in other *VAP*/*SCS* genes. Subsequently, sequence comparisons of the VAP/Scs proteins suggested that the presence of an additional downstream proline in VAPA (P63) and Scs2 (P58) might stabilize the MSP domain when the P56S mutation is present [38]. Indeed, the expression of VAPA^P56S, P63A^ in mammalian cells or Scs2^P51S, P58S^ in yeast resulted in the formation of ER inclusions, suggesting that humanizing the *SCS2* gene with an additional P58S substitution is necessary to observe ALS-like phenotypes due to the P51S mutation. Notably, Scs2^P51S, P58S^ inclusions were previously observed in *scs2*∆ cells expressing mutant protein from a high-copy (2µ) plasmid, suggesting the possibility of overexpression contributing to the altered distribution [38]. The heterologous overexpression in yeast of other aggregation-prone proteins linked to neurodegeneration, such as α-synuclein and a secretory pathway-targeted form of beta-amyloid (ssAβ_1–42_), results in toxic phenotypes or altered cellular behavior [119,120]. Since VAPB^P56S^ causes an autosomal dominant loss of function, we wanted to ensure that the phenotypes in our models occurred at single-copy expression levels. Thus, we generated chromosomal integration plasmids expressing *SCS2^WT^* or *scs2^P51S, P58S^* from the *P_SCS2_* promoter, which could be introduced into the genome at the *LEU2* gene locus. Strains modified with these plasmids expressed a single copy of wild-type or mutant *SCS2* from its endogenous promoter. Using this approach, we generated *scs2*∆ *scs22*∆ + *SCS2^WT^* and *scs2*∆ *scs22*∆ + *scs2^P51S, P58S^* cells to examine the effect of Scs2 re-expression on ER morphology and stress sensitivity (Figure 2). As controls, we generated WT and *scs2*∆ *scs22*∆ strains modified by the integration of an empty plasmid at the *LEU2* locus.

First, we examined the effect of *SCS2^WT^* and *scs2^P51S, P58S^* reintroduction on ER morphology in cells expressing Sec61-GFP from its endogenous gene locus. As shown in Figure 2A, WT cells had a normal ER morphology, with both nER and cER readily detected. As seen before (Figure 1A), *scs2*∆ *scs22*∆ cells with an empty integrating plasmid showed gaps in cER and the accumulation of Sec61-containing membrane structures in the cytoplasm, with brighter inclusion-like structures in the perinuclear region. The reintroduction of *SCS2^WT^* in *scs2*∆ *scs22*∆ cells resulted in an ER morphology similar to WT cells. In contrast, the expression of mutant scs2^P51S, P58S^ in *scs2*∆ *scs22*∆ cells did not fully restore cER in most cells and resulted in inclusion-like structures similar to those seen in *scs2*∆ *scs22*∆ cells. When we quantified the size of the inclusion-like structures as described above, we again found that they were significantly larger in *scs2*∆ *scs22*∆ cells than in WT controls (2.32-fold increase in median size), and this phenotype was corrected by the re-expression of *SCS2^WT^* from the *LEU2* locus (Figure 2B). The expression of scs2^P51S, P58S^ in *scs2*∆ *scs22*∆ cells failed to reduce the size of the inclusion-like structures compared to *scs2*∆ *scs22*∆ with an integrated empty vector. While there was a trend toward a further increase in size for *scs2*∆ *scs22*∆ + *scs2^P51S, P58S^* cells compared to *scs2*∆ *scs22*∆, this result failed to reach statistical significance. Regardless, the size of the inclusion-like structures in *scs2*∆ *scs22*∆ + *scs2^P51S, P58S^* cells remained significantly larger than in WT cells (3.00-fold increase in median size), suggesting that mutant scs2^P51S, P58S^ cannot restore the function of Scs2.

In addition to examining the ER morphology and inclusion-like structures, we also tested whether the re-expression of wild-type or mutant Scs2 could reduce the sensitivity to ER stress in *scs2*∆ *scs22*∆ cells lacking the Sec61-GFP tag. As in the previous experiment examining the effects of *scs2*∆, *scs22*∆ and *scs2*∆ *scs22*∆, we tested the growth of serially diluted strains on plates with increasing concentrations of tunicamycin. While *scs2*∆ *scs22*∆ cells showed weaker growth than WT cells on tunicamycin plates in a concentration-dependent manner, *scs2*∆ *scs22*∆ + *SCS2^WT^* grew similarly to WT cells at all drug concentrations (Figure 2C). In contrast, the re-expression of Scs2^P51S, P58S^ failed to rescue the tunicamycin sensitivity in *scs2*∆ *scs22*∆ cells, suggesting that the ALS8-mimicking mutant is also unable to correct ER stress sensitivity. Taken together, these results demonstrate that *scs2*∆ *scs22*∆ + *scs2^P51S, P58S^* cells phenocopy the ER morphology and stress sensitivity phenotypes seen in mammalian cells expressing VAPB^P56S^ and can serve as a fungal model for some aspects of ALS8.

### 3.3. Heterologous Expression of VAPB^WT^, but Not VAPB^P56S^, Complements ER Morphology and Stress Senstitivity in scs2∆ scs22∆ Cells

As an alternative yeast model for ALS8, we attempted to express human VAPB in *scs2*∆ *scs22*∆ yeast cells to test whether the human protein could complement the loss of SCS function in the yeast system. While earlier studies found that human VAPA shows similar localization to Scs2 when expressed heterologously in wild-type yeast cells [22], human VAPB has not been tested to our knowledge, and neither have been examined in the absence of endogenous Scs2 and Scs22. For these experiments, we cloned human *VAPB^WT^* and *VAPB^P56S^* into a low-copy yeast expression plasmid under the control of the *P_SCS2_* promoter and *T_SCS2_* terminator. We then subcloned the expression cassettes into the same integrating plasmids used for the introduction of wild-type and mutant *SCS2* at the *LEU2* locus and generated equivalent *scs2*∆ *scs22*∆ + *VAPB^WT^* and *scs2*∆ *scs22*∆ + *VAPB^P56S^* strains.

When we examined the localization of Sec61-GFP as an indicator of ER morphology, we found that the expression of human *VAPB^WT^* in *scs2*∆ *scs22*∆ cells could fully complement cER collapse and resulted in an ER morphology that appeared to be indistinguishable from that seen in WT cells (Figure 3A). In contrast, the heterologous expression of mutant *VAPB^P56S^* failed to rescue the cER collapse or formation of inclusion-like structures in *scs2*∆ *scs22*∆ cells. In agreement with our fluorescence microscopy data, the quantification of the inclusion-like structure size in each strain revealed that the heterologous expression of *VAPB^WT^* in *scs2*∆ *scs22*∆ cells significantly reduced their size compared to *scs2*∆ *scs22*∆ cells with an integrated empty vector (Figure 3B). In this regard, *scs2*∆ *scs22*∆ + *VAPB^WT^* cells were indistinguishable from WT controls, suggesting that human VAPB could functionally replace Scs2 in yeast. Unlike *VAPB^WT^*, the heterologous expression of *VAPB^P56S^* in *scs2*∆ *scs22*∆ cells failed to reduce the size of the inclusion-like structures and in fact resulted in a significant increase in their size compared to both WT cells and to *scs2*∆ *scs22*∆ cells with an integrated empty vector. Thus, VAPB^P56S^ could also recapitulate disease-like cellular phenotypes when expressed in our yeast ALS8 model.

In addition to examining the ER morphology, we tested whether the heterologous expression of human *VAPB^WT^* or *VAPB^P56S^* could reduce the sensitivity of *scs2*∆ *scs22*∆ cells to ER stress using the plate-based growth assay described above. In these experiments, the expression of *VAPB^WT^* improved the growth of *scs2*∆ *scs22*∆ cells at lower concentrations of tunicamycin (0.1875–0.25 µg/mL), but resulted in weaker than WT growth at higher concentrations (0.375–0.5 µg/mL; Figure 3C). In contrast, *scs2*∆ *scs22*∆ + *VAPB^P56S^* cells remained partially sensitive to tunicamycin, yielding slightly better growth than *scs2*∆ *scs22*∆ cells at lower concentrations of tunicamycin (0.1875–0.25 µg/mL), but slightly weaker growth than the equivalent VAPB^WT^-expressing cells across the entire range of drug concentrations. Overall, these data suggest that wild-type, but not mutant, VAPB can complement the ER morphology defects that result from the loss of *SCS* genes in yeast, but that VAPB is less effective than Scs2 in complementing ER stress sensitivity.

### 3.4. Multiple ER-Resident Proteins Localize to Inclusion-like Structures in SCS-Deficient Cells

Previous studies suggested that multiple proteins may be recruited into ER inclusions in mammalian cells expressing mutant VAPB^P56S^. Our analysis of the inclusion-like structures in the yeast ALS8 models was thus far restricted to examining Sec61, a transmembrane protein in the translocon complex. To assess whether other proteins were similarly found in inclusion-like structures, we examined the localization of two additional ER-resident cargos: the reticulon Rtn1, which is a largely cytoplasmic-facing protein that inserts into the ER membrane via hydrophobic loops, and DsRed-HDEL, a soluble luminal cargo [43,121,122]. We expressed each of these cargos in WT, *scs2*∆, *scs22*∆ and *scs2*∆ *scs22*∆ cells, as well as *scs2*∆ *scs22*∆ cells with wild-type or mutant *SCS2* or *VAPB* reintroduced as described above. In all cases, cells also expressed Sec61-GFP from its chromosomal locus to determine whether Rtn1-mScarlet1 (tagged at its chromosomal locus) or DsRed-HDEL (expressed from a low-copy plasmid) localized to the same inclusion-like structures that we observed upon loss of *SCS* function. As shown in Figure 4A,B, both Rtn1-mScarlet1 and DsRed-HDEL prominently co-localized with Sec61-GFP under all conditions, and these additional cargos were readily observed in inclusion-like structures containing Sec61-GFP. Thus, multiple ER-resident proteins with differing topologies localize to aberrant inclusion-like structures.

### 3.5. Kinetic Growth Assays Reveal Differences in Tunicamycin Sensitivity in Yeast ALS8 Model Cells

A noted limitation of our plate-based growth assays is that they are qualitative rather than quantitative. Moreover, the timing of data collection can create challenges in the interpretation of the results, since slower-growing strains eventually catch up with faster-growing strains on a plate, thereby masking actual differences in growth. As an additional method to assess the sensitivity to ER stress, we developed a plate reader-based assay for the simultaneous monitoring of the growth kinetics in WT, *scs2*∆ *scs22*∆ and *scs2*∆ *scs22*∆ + wild-type or mutant *SCS2* or *VAPB* strains with increasing concentrations of tunicamycin. With these kinetic assays, we were able to compare the relative growth of each strain with increasing drug concentrations.

Analysis of the kinetic assays revealed two observable differences in growth behavior for the cells tested (Figure 5A–F). First, the growth curves in the absence of the drug differed between strains, where *scs2*∆ *scs22*∆, *scs2*∆ *scs22*∆ + *scs2^P51S, P58S^* and *scs2*∆ *scs22*∆ + *VAPB^P56S^* strains appeared to show a significant reduction in growth compared to strains with a functional Scs/VAP-family protein (WT, *scs2*∆ *scs22*∆ + *SCS2^WT^* or *scs2*∆ *scs22*∆ + *VAPB^WT^*). This difference was particularly clear when we compared strains at a time point where cells were still in their logarithmic growth phase and had not reached a plateau in growth, where slower-growing cultures could start to catch up in density with faster-growing strains. The 12 h time point of our growth curve experiments reflected a late logarithmic phase; thus, we used this time point for all comparisons of relative growth. Based on this analysis, we found that *scs2*∆ *scs22*∆, *scs2*∆ *scs22*∆ + *scs2^P51S, P58S^* and *scs2*∆ *scs22*∆ + *VAPB^P56S^* cells showed reduced growth compared to WT cells in the absence of tunicamycin (Figure 5G). In contrast, the growth of *scs2*∆ *scs22*∆ + *SCS2^WT^* and *scs2*∆ *scs22*∆ + *VAPB^WT^* cells was indistinguishable from WT and significantly improved compared to *scs2*∆ *scs22*∆. The growth of *scs2*∆ *scs22*∆ + *VAPB^P56S^* cells was slightly reduced compared to *scs2*∆ *scs22*∆ cells, suggesting that the expression of the human ALS8-linked allele may have a negative impact on growth in these cells.

In addition to differences in baseline growth without the drug, each strain showed a dose-dependent reduction in growth in response to tunicamycin (Figure 5A–F). Using the 12 h time point in our kinetic assays as a reference, 0.5 µg/mL tunicamycin resulted in an approximately 50% reduction in the growth of WT cells compared to no drug. Consequently, we compared the relative growth of each strain by normalizing the growth in 0.5 µg/mL tunicamycin to that of the same strain’s grown in the absence of the drug (Figure 5H). We found that *scs2*∆ *scs22*∆ cells showed significantly weaker growth in the presence of 0.5 µg/mL tunicamycin than WT cells. The re-expression of *SCS2^WT^* in *scs2*∆ *scs22*∆ cells restored growth to levels that were indistinguishable from WT cells and significantly greater than *scs2*∆ *scs22*∆ cells with an integrated empty vector. In contrast, *scs2*∆ *scs22*∆ cells expressing scs2^P51S, P58S^ grew significantly weaker than WT cells and were statistically indistinguishable from *scs2*∆ *scs22*∆ cells with an empty vector. The relative growth of *scs2*∆ *scs22*∆ + *VAPB^WT^* and cells in 0.5 µg/mL tunicamycin was statistically indistinguishable from both WT and *scs2*∆ *scs22*∆ cells, while the growth of *scs2*∆ *scs22*∆ + *VAPB^P56S^* cells was similar to WT and significantly improved compared to *scs2*∆ *scs22*∆ cells. Based on these results, our yeast ALS8 model expressing mutant *scs2* appears to be more sensitive to tunicamycin than the equivalent model expressing mutant *VAPB*.

With the data from the kinetic assays, we also generated dose–response curves (Figure 5I) and used the best-fit curve from the average of five independent trials based on their 12 h growth to estimate the IC_50_ of each strain, where curves reflected normalized growth relative to no drug. This analysis yielded IC_50_ values of 0.578 ± 0.035 µM (WT), 0.440 ± 0.026 µM (*scs2*∆ *scs22*∆), 0.726 ± 0.041 µM (*scs2*∆ *scs22*∆ + *SCS2^WT^*), 0.501 ± 0.022 µM (*scs2*∆ *scs22*∆ + *scs2^P51S, P58S^*), 0.6576 ± 0.059 µM (*scs2*∆ *scs22*∆ + *VAPB^WT^*) and 0.566 ± 0.049 µM (*scs2*∆ *scs22*∆ + *VAPB^P56S^*). Taken together, these data suggest that the re-expression of wild-type, but not ALS8 mutant, Scs2 restores the tunicamycin resistance of *scs2*∆ *scs22*∆ cells to WT levels in kinetic assays as well as in plate-based growth assays. Although human VAPB was able to correct cER collapse in *scs2*∆ *scs22*∆ cells, it could not fully complement ER stress sensitivity.

### 3.6. Potential Use of Yeast as a Model for Study of Other Types of ALS

While we generated yeast models for ALS8 in this study, ALS is a complex disease with mutations in over 30 genes linked to familial cases. To expand the potential use of yeast for the study of other forms of ALS, we examined all human genes currently linked to ALS based on annotation in Online Mendelian Inheritance in Man (OMIM) and identified a total of 17 ALS-linked genes in humans with yeast homologs. We then performed pair-wise sequence alignments between the human and yeast proteins to identify disease-linked mutations in conserved or similar amino acids (Table 3 and Appendix A). These may provide useful avenues for the generation of models of other ALS subtypes to study the cellular effects of disease-causing mutations or for the design and execution of genetic screens to identify phenotypic suppressors. Notably, mutations in some ALS-linked genes are also associated with other diseases. We also note these in Table 3, as our analysis could create opportunities to study diseases beyond ALS.

## 4. Discussion

### 4.1. Yeast as a Model System for Study of Neurodegenerative Diseases

Yeast have a long history of yielding important insights into fundamental cellular processes that are conserved through eukaryotic evolution. Given their rapid growth and ease of genetic manipulation, yeast have proven to be especially useful in performing screens to identify the set of genes required for basic cellular processes, including cell cycle regulation, secretion and endocytosis. More recently, yeast have been utilized as model systems for the study of human diseases, including neurodegeneration. Several key studies have shown that the ectopic expression of human proteins linked to neurodegeneration results in disease-like phenotypes in yeast, and these have been used to study the cellular basis of toxicity. These include the expression of α-synuclein (α-Syn) and β-amyloid (Aβ_1–42_), which respectively are linked to Parkinson’s disease and Alzheimer’s disease [119,120]. Although yeast are unicellular organisms that do not possess many features seen in neuronal cells, the observation that the heterologous expression of human disease-linked proteins can result in toxicity suggests that these proteins are involved in basic cellular functions that are broadly conserved in eukaryotes. Importantly, these yeast models have been used in genetic screens to identify genes and protein products that can suppress toxic or disease-related phenotypes. Such studies provide critical insights into the molecular basis of neurodegeneration, as well as the network of proteins and cellular pathways that interact with or are affected by proteins implicated in neurodegenerative diseases. Moreover, these studies can serve as a starting point for the identification of conserved targets for drug development. Thus, yeast represent a powerful discovery platform for the understanding of eukaryotic cell function, both in health and in disease.

### 4.2. Yeast ALS8 Models May Facilitate Studies of VAPB^P56S^-Related Cellular Pathology

In this study, we generated two yeast models for ALS8 using *scs2*∆ *scs22*∆ cells lacking all endogenous yeast VAPs as a starting point. Although *scs2*∆ alone caused the partial loss of cER and resulted in sensitivity to the ER stress-inducing drug tunicamycin, both phenotypes were exacerbated in *scs2*∆ *scs22*∆ double mutant cells. These findings suggest that Scs2 and Scs22 share at least partially overlapping functions; thus, we chose to delete both genes in our model system to avoid the possibility of compensation by Scs22.

While a previous study found that *SCS2^P51S^*, which is analogous to *VAPB^P56S^*, did not cause a loss-of-function phenotype [22], later work demonstrated that the humanization of *SCS2* by mutating a non-conserved proline residue (Pro58) in proximity to Pro51 resulted in Scs2 inclusions and inositol auxotrophy when expressed from a high-copy plasmid in *scs2*∆ cells [38]. Multicopy plasmids with a 2µ origin of replication are typically used to overexpress proteins, and cell-to-cell differences in plasmid copy number can introduce variability that may impact the efficacy and usability of the model. To ensure that our models were as consistent as possible, we integrated a single copy of WT or mutant *SCS2* or human *VAPB* into the yeast genome. Each gene was expressed from the native *P_SCS2_* promoter to achieve protein expression as close to endogenous levels as possible and to ensure that our models more accurately reflected protein levels that are relevant in the context of normal cellular function.

The first model that we generated was an *scs2*∆ *scs22*∆ strain in which we re-expressed *SCS2^WT^* as a control or *scs2^P51S, P58S^* from its own promoter, at an alternative gene locus (*LEU2*), as the yeast equivalent of human *VAPB^P56S^*. In the second model, we used the same starting strain but instead integrated human *VAPB^WT^* or *VAPB^P56S^* for exogenous expression at the *LEU2* gene locus. Both models gave similar overall results in terms of the WT alleles restoring cER, reducing the size of Sec61-containing inclusion-like structures and fully (SCS2^WT^) or partially (VAPB^WT^) reducing sensitivity to ER stress, as assessed by growth on plates or in liquid culture in the presence of tunicamycin. In contrast, neither Scs2^P51S, P58S^ nor VAPB^P56S^ restored cER, and Scs2^P51S, P58S^ also failed to correct the sensitivity to ER stress. The expression of VAPB^P56S^ in *scs2*∆ *scs22*∆ cells at least partially improved growth in the presence of tunicamycin (Figure 5H); although the underlying reason for this is currently unclear, we speculate that the altered interaction between human VAPB and the yeast homologs of its binding partners may be a contributing factor. While Sec61-GFP inclusions in *scs2*∆ *scs22*∆ + *scs2^P51S, P58S^* were clearly larger than in WT controls, they were not statistically different from *scs2*∆ *scs22*∆ cells, even though the median inclusion area trended toward being larger in cells expressing the mutant protein. Thus, as suggested for *VAPB^P56S^*, *scs2^P51S, P58S^* may represent a true loss-of-function allele, where the expression of the mutant protein in *scs2*∆ *scs22*∆ cells is phenotypically not discernable from cells lacking *SCS* function altogether [13,31]. Similarly, the expression of *VAPB^P56S^* in *scs2*∆ *scs22*∆ failed to correct the ER morphology and stress sensitivity defects and in fact resulted in a significant increase in the median size of the inclusion-like structures compared to *scs2*∆ *scs22*∆ cells. ALS8-linked mutant VAPB may therefore also represent a loss of function when expressed in yeast, or could alternatively adopt a toxic gain of function with respect to ER morphology.

We envision that these two yeast models will enable studies of the cellular mechanisms underlying disease phenotypes due to VAPB^P56S^ expression in several ways, especially since we were able to phenocopy some aspects of mammalian cell culture models of ALS8 [18,29,31]. Specifically, they could be subjected to genetic screens to identify enhancers or suppressors of ALS8-related phenotypes, either by the overexpression or by the deletion of potential modifiers, using phenotypes such as tunicamycin sensitivity or inositol auxotrophy [22,24,38,123]. Similar screens have previously been conducted using yeast models that exogenously express α-Syn or Aβ_1–42_, which are causative for Parkinson’s and Alzheimer’s diseases, respectively [119,120]. These screens relied on the high-copy expression of yeast genes to correct the toxic effects of a non-conserved gene, as yeast do not possess homologs of α-Syn or Aβ_1–42_ or its precursor protein. Since *SCS2* is the yeast homolog of *VAPB*, it may be advantageous to perform genetic screens using the model expressing *scs2^P51S, P58S^*, since its expression is more likely to conserve the physical interactions with other yeast proteins, including those overexpressed in a genetic screen [124]. As the results from a screen are validated, our observation that human VAPB can substitute the function of yeast Scs2 might be used to test whether candidate suppressor or enhancer genes also impact phenotypes resulting from the expression of the human protein. Of particular interest from this approach, genes that are conserved through evolution can be further studied in cell culture or organismal models of ALS8 to determine whether mammalian homologs can similarly reduce toxic phenotypes.

Aside from classical genetic screens, these yeast models can also be used for chemical genetic screens aimed at identifying pharmaceuticals or small molecules that ameliorate stress sensitivity and/or ER morphology defects. If the goal of these screens is to identify drugs that directly affect the function, folding or stability of mutant Scs2/VAPB, *scs2*∆ *scs22*∆ cells expressing *VAPB^P56S^* might be the more attractive model, since there is a higher likelihood of obtaining effective lead compounds for human therapeutic use compared to drug screens against yeast Scs2 [124].

### 4.3. Similarities and Differences between ALS8 Models

In addition to the yeast system described here, other model organisms have been used to study ALS8 with varying outcomes. Several transgenic mouse models have been generated, where *VAPB*^−/−^ mice developed mild motor deficits at 12 months in one study, while the overexpression of *VAPB^WT^* or *VAPB^P56S^* caused protein inclusions but did not result in obvious behavioral or neurological symptoms consistent with disease [125,126]. In one of these studies, the knockdown of *VAPB* in zebrafish was achieved using antisense morpholinos and resulted in moderate to severe motor deficits that were complemented by the expression of human *VAPB^WT^*, but not *VAPB^P56S^* [126]. In *Caenorhabditis elegans*, loss of the *VAPB* homolog *vpr-1* caused locomotor defects and motor neuron death, while the ectopic expression of human *VAPB^P56S^* caused axon guidance defects [127]. *Drosophila melanogaster* models of ALS8 have also been generated, demonstrating that a proteolytic cleavage product of VAPB is secreted and acts as a trophic factor through interaction with Eph receptors and that mutant VAPB is less efficiently processed and secreted [33]. The expression of *Drosophila VAP33A^P58S^* results in synaptic defects reminiscent of a loss of function and causes neuronal death, VAP-containing inclusions and locomotor defects in fly larvae [32,128]. Together, these models provide a variety of opportunities to study the effects of VAPB mutation at the cellular and organismal level. While they may differ from each other and from mammalian cell culture models or human patients in some respects, they are nonetheless important tools in increasing our understanding of neurodegeneration in ALS8. Notably, most forms of ALS are late-onset, although juvenile cases have also been documented. While the lifespan of most models used to study ALS is significantly shorter than the typical age of onset in humans (>50 years), the ability to observe disease-related phenotypes in these systems can provide key insights into the mechanisms of neuronal loss that may arise from perturbations in basic functions that are shared across a wide variety of cell types. We speculate that damage due to mild or moderate cellular stresses may accrue over time at the organismal level, and this may contribute to later-onset symptoms that are seen in humans but might not be as readily observed in shorter-lived organisms under normal circumstances.

### 4.4. Yeast May Provide Opportunities and Approaches for Study of Other ALS Subtypes

Although the present study focused on generating fungal models of VAPB/ALS8 as a subtype of amyotrophic lateral sclerosis, yeast may serve as a rich source of information for the understanding and modeling of other types of ALS. As summarized in Table 3, many genes linked to ALS in humans are conserved through evolution and have yeast homologs. We performed sequence alignments of human ALS-linked proteins and yeast orthologs and were able to identify many mutations that affect conserved or similar amino acids in both organisms. If the mutation or loss of function of the yeast gene yields an observable phenotype, it may be possible to determine whether the human homolog can replace the function of the yeast gene and to test whether disease-linked mutations also result in phenotypes that can be studied in the context of a disease. Some of the yeast genes are essential, either as single genes or in groups where the loss of function results in inviable cells (Table 3, noted in red or blue, respectively); these might be particularly useful for the study of mutant alleles, as they can quickly yield information about whether a mutation causes a loss of function. In several cases, the yeast homolog of a human gene involved in ALS did not have conserved or similar amino acid residues corresponding to the disease-linked mutations. Ectopic replacement with the wild-type or mutant human gene may thus be a more pragmatic approach to the generation of yeast models of ALS in these instances. Since the ectopic expression of α-Syn and Aβ_1–42_ yields toxic phenotypes in yeast [119,120], it may also be possible to express human ALS-linked genes that are not conserved, to study their phenotypes in a similar manner. It should be noted that most mutations in the yeast homologs described in Table 3 have not yet been generated and tested for phenotypes relevant to ALS and may not necessarily yield useful disease models; the listed alleles should thus be considered a starting point for future experiments.

As we examined these conserved ALS-linked genes for similarities between yeast and human orthologs, we noted that several of the human ALS-linked genes have also been implicated in other diseases, many of which include neurological symptoms. Thus, it may be possible to adapt our approach to generate tractable models of other neurodegenerative or neurological disorders.

## 5. Conclusions

Overall, we have developed yeast models of ALS8 that express either mutant human VAPB^P56S^ or the yeast homolog Scs2^P51S, P58S^ and found that our model displays phenotypes that are reminiscent of those seen in mammalian cells expressing VAPB^P56S^. We propose that these yeast models can be used to study basic aspects of cellular dysfunction in disease that may yield important insights into ALS8 in human cells and will provide opportunities for the identification of therapeutic targets. Many ALS-linked genes are conserved through evolution; thus, our approach will likely be adaptable to other genetic causes of ALS, as well as to other human diseases.

## Figures and Tables

**Figure 1 biomolecules-13-01147-f001:**
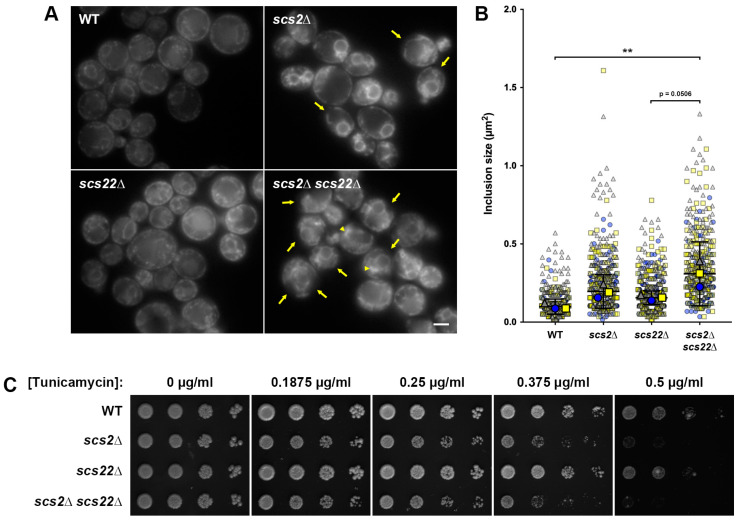
Effect of *SCS2* and *SCS22* deletion on ER morphology and stress sensitivity. (**A**) WT, *scs2*∆, *scs22*∆ and *scs2*∆ *scs22*∆ cells expressing the ER marker Sec61-GFP from its endogenous locus were imaged by fluorescence microscopy. Scale bar, 2.5 µm. (**B**) Quantification of the size of the largest inclusion-like structure per cell from images collected in experiments from panel (**A**). SuperPlots display individual measurements color- and shape-coded in the background (blue circles, yellow squares, gray triangles) for three independent trials (*n* > 100 cells measured per trial). Median values for each trial are shown in the foreground, with bars corresponding to mean ± 95% confidence interval based on trial medians; ** *p* < 0.01. (**C**) Serial dilutions of WT, *scs2*∆, *scs22*∆ and *scs2*∆ *scs22*∆ cells grown to mid-logarithmic phase were spotted onto YNB plates with increasing tunicamycin concentrations as indicated and grown for 4 days prior to imaging.

**Figure 2 biomolecules-13-01147-f002:**
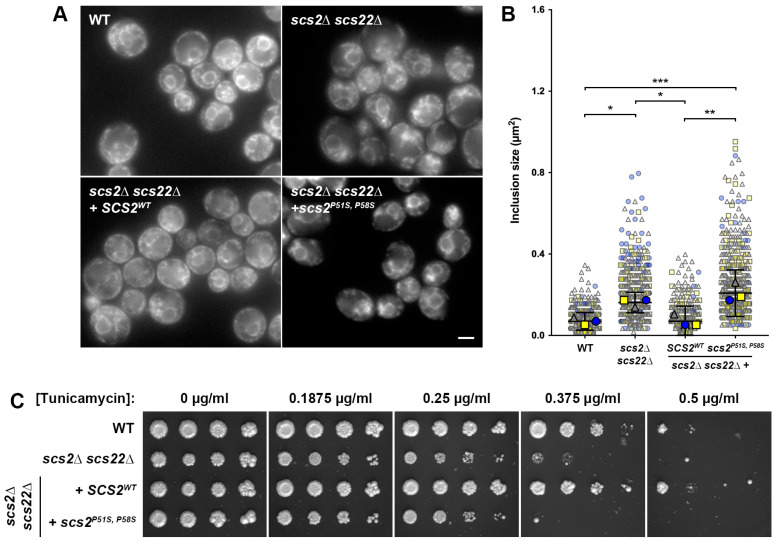
Effect of *SCS2*^WT^ and *scs2^P51S, P58S^* expression on ER morphology and stress sensitivity in *scs2*∆ *scs22*∆ cells. (**A**) WT, *scs2*∆ *scs22*∆ and *scs2*∆ *scs22*∆ + *SCS2^WT^* or *scs2^P51S, P58S^* cells expressing Sec61-GFP from its endogenous locus were imaged by fluorescence microscopy. Scale bar, 2.5 µm. (**B**) Quantification of the size of the largest inclusion-like structure per cell from images collected in experiments from panel (**A**). SuperPlots display individual measurements color- and shape-coded in the background (blue circles, yellow squares, gray triangles) for three independent trials (*n* > 100 cells measured per trial). Median values for each trial are shown in the foreground, with bars corresponding to mean ± 95% confidence interval based on trial medians; * *p* < 0.05, ** *p* < 0.01, *** *p* < 0.001. (**C**) Serial dilutions of WT, *scs2*∆ *scs22*∆ and *scs2*∆ *scs22*∆ + *SCS2^WT^* or *scs2^P51S, P58S^* cells grown to mid-logarithmic phase were spotted onto YNB plates with increasing tunicamycin concentrations as indicated and grown for 6 days prior to imaging.

**Figure 3 biomolecules-13-01147-f003:**
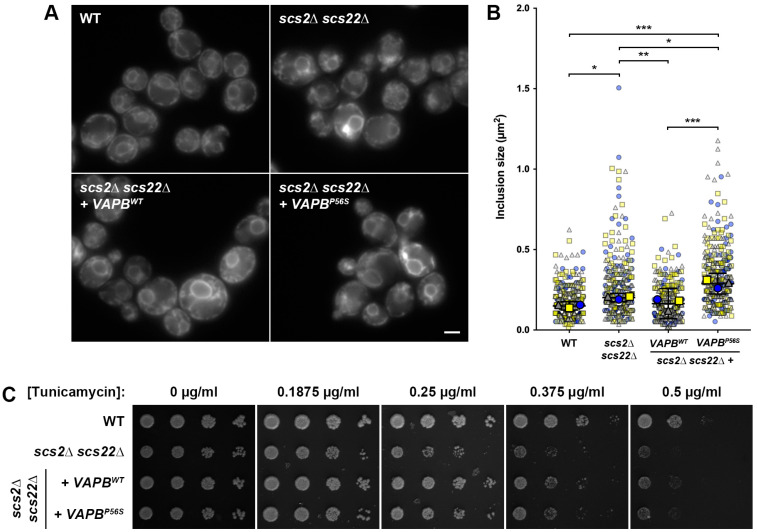
Effect of *VAPB*^WT^ and *VAPB^P56S^* expression on ER morphology and stress sensitivity in *scs2*∆ *scs22*∆ cells. (**A**) WT, *scs2*∆ *scs22*∆ and *scs2*∆ *scs22*∆ + *VAPB^WT^* or *VAPB^P56S^* cells expressing Sec61-GFP from its endogenous locus were imaged by fluorescence microscopy. Scale bar, 2.5 µm. (**B**) Quantification of the size of the largest inclusion-like structure per cell from images collected in experiments from panel (**A**). SuperPlots display individual measurements color- and shape-coded in the background (blue circles, yellow squares, gray triangles) for three independent trials (*n* > 100 cells measured per trial). Median values for each trial are shown in the foreground, with bars corresponding to mean ± 95% confidence interval based on trial medians; * *p* < 0.05, ** *p* < 0.01, *** *p* < 0.001. (**C**) Serial dilutions of WT, *scs2*∆ *scs22*∆ and *scs2*∆ *scs22*∆ + *VAPB^WT^* or *VAPB^P56S^* cells grown to mid-logarithmic phase were spotted onto YNB plates with increasing tunicamycin concentrations as indicated and grown for 4 days prior to imaging.

**Figure 4 biomolecules-13-01147-f004:**
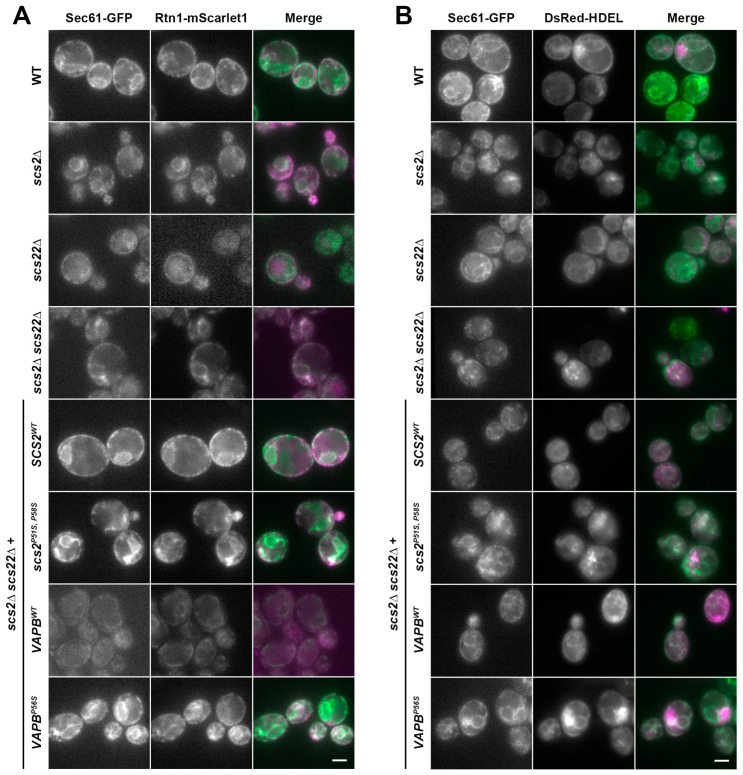
Co-localization of Sec61-GFP with other ER markers in inclusion-like structures. (**A**) WT, *scs2*∆, *scs22*∆, *scs2*∆ *scs22*∆, as well as *scs2*∆ *scs22*∆ cells with *SCS2^WT^*, *scs2^P51S, P58S^*, *VAPB^WT^* or *VAPB^P56S^* as indicated, were examined by fluorescence microscopy to assess co-localization of Sec61-GFP (green) with Rtn1-mScarlet1 (magenta). Both genes were C-terminally tagged at their endogenous chromosomal loci. (**B**) The same set of cells expressing DsRed-HDEL (magenta) from a low-copy plasmid instead of Rtn1-mScarlet1 was also imaged by fluorescence microscopy. Scale bar, 2.5 µm.

**Figure 5 biomolecules-13-01147-f005:**
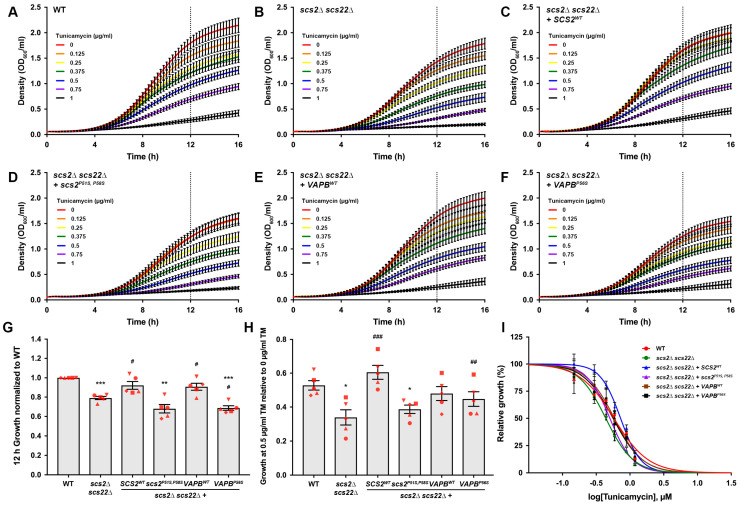
Kinetic growth assays for assessment of relative growth and ER stress sensitivity in WT and ALS8 model yeast cells. The 16 h growth curves of (**A**) WT, (**B**) *scs2*∆ *scs22*∆, (**C**) *scs2*∆ *scs22*∆ + *SCS2^WT^*, (**D**) *scs2*∆ *scs22*∆ + *scs2^P51S, P58S^*, (**E**) *scs2*∆ *scs22*∆ + *VAPB^WT^* or (**F**) *scs2*∆ *scs22*∆ + *VAPB^P56S^* cells grown in the absence or presence of tunicamycin at the indicated concentrations. Curves represent the mean ± s.e.m., where independent trials for each strain in panels (**A**–**F**) were conducted simultaneously (*n* = 5). Dashed line at 12 h indicates the time used for comparisons of relative growth. (**G**) Relative growth at 12 h in the absence of tunicamycin for each strain from panels (**A**–**F**) normalized to WT; bars correspond to mean ± s.e.m. (*n* = 5). (**H**) Relative growth at 12 h in the presence of 0.5 µg/mL tunicamycin for strains from panels (**A**–**F**) normalized to cells grown in the absence of drug; bars correspond to mean ± s.e.m. (*n* = 5). (**I**) Dose–response curves generated from experiments in panels (**A**−**F**) (*n* = 5). Growth at 12 h for each drug concentration was normalized to cells grown in the absence of drug. In panels (**G**,**H**), values for paired data points (marked in red) are matched by shape across all conditions. All error bars correspond to mean ± s.e.m. * *p* < 0.05, ** *p* < 0.01, *** *p* < 0.001, compared to WT; ^#^ *p* < 0.05, ^##^
*p* < 0.01, ^###^ *p* < 0.001 compared to *scs2*∆ *scs22*∆.

**Table 1 biomolecules-13-01147-t001:** Yeast strains used in this study.

Strain	Genotype	Source
W303 ^1^	*MATa leu2-3,112 trp1-1 can1-100 ura3-1 his3-11,15*	Laboratory strain
DPY2271	*MATa leu2-3,112::LEU2*	This study
DPY2275	*MATa leu2-3,112::LEU2 Sec61-GFP::HIS3*	This study
DPY2287	*MATa scs2::KANMX6 leu2-3,112::LEU2 Sec61-GFP::HIS3*	This study
DPY2290	*MATa scs22::HPHMX4 leu2-3,112::LEU2*	This study
DPY2295	*MATa scs22::HPHMX4 leu2-3,112::LEU2 Sec61-GFP::HIS3*	This study
DPY2298	*MATa scs2::KANMX6 scs22::HPHMX4 leu2-3,112::LEU2*	This study
DPY2301	*MATa scs2::KANMX6 scs22::HPHMX4 leu2-3,112::LEU2 Sec61-GFP::HIS3*	This study
DPY2309	*MATa scs2::KANMX6 scs22::HPHMX4 leu2:SCS2::LEU2*	This study
DPY2311	*MATa scs2::KANMX6 scs22::HPHMX4 leu2:SCS2::LEU2 Sec61-GFP::HIS3*	This study
DPY2317	*MATa scs2::KANMX6 scs22::HPHMX4 leu2:scs2^P51S, P58S^::LEU2*	This study
DPY2319	*MATa scs2::KANMX6 scs22::HPHMX4 leu2:scs2^P51S, P58S^::LEU2 Sec61-GFP::HIS3*	This study
DPY2339	*MATa scs2::KANMX6 leu2-3,112::LEU2*	This study
DPY2822	*MATa scs2::KANMX6 scs22::HPHMX4 leu2:hsVAPB::LEU2 Sec61-GFP::HIS3*	This study
DPY2823	*MATa scs2::KANMX6 scs22::HPHMX4 leu2:hsVAPB^P56S^::LEU2 Sec61-GFP::HIS3*	This study
DPY2854	*MATa scs2::KANMX6 scs22::HPHMX4 leu2:hsVAPB::LEU2*	This study
DPY2855	*MATa scs2::KANMX6 scs22::HPHMX4 leu2:hsVAPB^P56S^::LEU2*	This study
DPY2952	*MATa leu2-3,112::LEU2 Sec61-GFP::HIS3 Rtn1-mScarlet1::NATMX4*	This study
DPY2953	*MATa scs2::KANMX6 leu2-3,112::LEU2 Sec61-GFP::HIS3 Rtn1-mScarlet1::NATMX4*	This study
DPY2954	*MATa scs22::HPHMX4 leu2-3,112::LEU2 Sec61-GFP::HIS3 Rtn1-mScarlet1::NATMX4*	This study
DPY2955	*MATa scs2::KANMX6 scs22::HPHMX4 leu2-3,112::LEU2 Sec61-GFP::HIS3 Rtn1-mScarlet1::NATMX4*	This study
DPY2956	*MATa scs2::KANMX6 scs22::HPHMX4 leu2:SCS2::LEU2 Sec61-GFP::HIS3 Rtn1-mScarlet1::NATMX4*	This study
DPY2957	*MATa scs2::KANMX6 scs22::HPHMX4 leu2:scs2^P51S, P58S^::LEU2 Sec61-GFP::HIS3 Rtn1-mScarlet1::NATMX4*	This study
DPY2958	*MATa scs2::KANMX6 scs22::HPHMX4 leu2:hsVAPB::LEU2 Sec61-GFP::HIS3 Rtn1-mScarlet1::NATMX4*	This study
DPY2959	*MATa scs2::KANMX6 scs22::HPHMX4 leu2:hsVAPB^P56S^::LEU2 Sec61-GFP::HIS3 Rtn1-mScarlet1::NATMX4*	This study

^1^ All strains used in this study were derived from W303 as the parental wild-type strain.

**Table 2 biomolecules-13-01147-t002:** Plasmids used in this study.

Plasmid	Description	Source
pRS405	*LEU2* integrating plasmid	[42]
pRS413	*CEN HIS3* empty vector	[42]
pRS416	*CEN URA3* empty vector	[42]
pDP0558	*SCS2*.413 [*CEN HIS3*]	This study
pDP0559	*scs2^P51S, P58S^*.413 [*CEN HIS3*]	This study
pDP0562	*SCS2*.405 [*LEU2*]	This study
pDP0563	*scs2^P51S, P58S^*.405 [*LEU2*]	This study
pDP0900	*P_SCS2_-hsVAPB^WT^-T_SCS2_*.416 [*CEN URA3*]	This study
pDP0901	*P_SCS2_-hsVAPB^P56S^-T_SCS2_*.416 [*CEN URA3*]	This study
pDP0911	*P_SCS2_-hsVAPB^WT^-T_SCS2_*.405 [*LEU2*]	This study
pDP0912	*P_SCS2_-hsVAPB^P56SS^-T_SCS2_*.405 [*LEU2*]	This study
pHDEL	pRS416.DsRed-HDEL [*CEN URA3*]	[43]

**Table 3 biomolecules-13-01147-t003:** List of ALS-linked genes conserved between yeast and human, with conserved disease-linked mutations (essential yeast genes noted in red; essential combinations noted in blue).

ALS Type	Human Gene (Accession)	Yeast Homolog	Identity (%) ^a^	Human Mutation (Yeast Analog) ^a^	Suppl. File	Refs.
ALS1	*SOD1* (NP_000445.1)	*SOD1*	54.9	A4T/V(A4); C6F(A6); G12R(A12); G16S(G16); E21K(E21); G37R(G37); G41D/S(A42); H43R(R44); F45C(F46); H46R(H47); G72S(G73); H80R(H81); L84V(M85); G85R(G86); D90A(D91); G93A/C/R(G94); D95N(K97); E100G(K101); I104F(I105); L106V(L107); I113T(V114); V119∆ or V120∆(V118 or V119) ^b^; L126T(L127); S134N(S135); A145T(A146); I151T(L152)	2-1	[45,46,47,48,49,50,51,52,53,54,55,56,57,58,59,60,61,62,63,64,65,66]
ALS4	*SETX* (AAI37351.1)	* SEN1 *	23.1	No conserved ALS-linked mutations SCA-linked ^c^: M271I(I137); E343*(K230) ^d^; Q653K(Q404); Q868*(N587); R1363*(N895); L1976R(I1370); L1977F(I1371); P2213L(P1622)	2-2	[67,68,69,70,71,72]
ALS8	*VAPB* (NP_004729.1)	*SCS2* *SCS22*	26.34 28.57	T46I(T41); P56S(P51) T46I(T38); P56S(P48)	2-3	[28,73]
ALS11	*FIG4* (NP055660.1)	*FIG4*	38.03	D53Y(E72); R183*(K195) BTOP-linked ^e^: D783V(H776) CMT4J-linked ^f^: L17P(L35); I41T(I59); F98fsTer102(F109); G253fsTer261(G244); E302K(E293) YVS-linked ^g^: L175P(L187); K278WfsTer6(H269); T422NfsTer6(S422)	2-4	[74,75,76,77,78,79]
ALS13	*ATXN2* (AAB19200.1)	*PBP1*	24.06	No conserved ALS-linked mutations; (CAG)_n_ repeat expansion (*n* = 27–33 in ALS13)	2-5	[80]
ALS14/ FTD-ALS6	*VCP* (AAI10914.1)	* CDC48 *	69.48	R155H/C(R165); R159G/S(R169); D395G(D405); D592N(D602) IBMPFD1-linked ^h^: N91Y(N101); R95G(R105); R155P(R165); R159H(R169); R191Q(R201); A232E(A242) CMT2Y-linked ^i^: G97E(G107); E185K(E195)	2-6	[81,82,83,84,85,86,87,88]
ALS15	*UBQLN2* (NP_038472.2)	*DSK2*	30.31	No conserved ALS-linked mutations	2-7	
ALS16	*SIGMAR1* (NP_005857.1)	*ERG2*	31.6	E102Q(E104)	2-8	[89]
ALS17/ FTD-ALS7	*CHMP2B* (NP_054762.2)	*DID4*	28.17	D148Y(N152); Q165*(K170); Q206H(R224)	2-9	[90,91,92]
ALS18	*PFN1* (AAA36486.1)	* PFY1 *	29.03	No conserved ALS-linked mutations	2-10	
ALS20	*HNRNPA1* (NP_112420.1)	* HRP1 *	33.33	D314N(D490); N319S(N497) IBMPFD3-linked ^j^: D314V(D490)	2-11	[93]
ALS22	*TUBA4A* (AAH09238.1)	* TUB1 * *TUB3*	74.44 72.3	T145P(T146); R215C(R216); R320C/H(R321); A383T(S384); W407*(W408) Macrothrombocytopenia: V181M(V182), E183Q(E184) T145P(T146); R215C(R216); R320C/H(R321); A383T(S384); W407*(W408) Macrothrombocytopenia: V181M(V182), E183Q(E184)	2-12	[94,95]
ALS25	*KIF5A* (AAA20231.1)	*SMY1*	26.49	No conserved ALS-linked mutations Spastic paraplegia 10: R204Q(R234); D232N(D262); G235E(G265); N256S(N286); R280C/H(R319)	2-13	[96,97,98,99,100]
ALS26	*TIA1* (NP_071505.2)	*PUB1*	34.82	No conserved ALS-linked mutations	2-14	
ALS27	*SPTLC1* (NP_006406.1)	* LCB1 *	35.32	Y23F(Y55); F40∆, S41∆ (L72, S73); S331Y(G378) HSAN1A ^k^: C133Y/W(C180); V144D(V191); S331F(G379); A352V(V399)	2-15	[101,102,103,104,105]
FTD-ALS2	*CHCHD10* (AAH65232.1)	*MIX17*	35.25	S59L(S70) SMAJ-linked ^l^: G66V(G77) IMMD-linked ^m^: G58R(G69)	2-16	[106,107,108]
FTD-ALS5	*CCNF* (AAH12349.1)	* CLN1 * * CLN2 * * CLN3 *	18.14 19.66 22.75	R392T(Q337) R392T(H136) S195R(A2); S621G(S423)	2-17	[108]

^a^ % identity and analogous amino acid residues were determined based on alignments using Clustal Omega; ^b^ ∆ signifies an in-frame deletion; ^c^ SCA: spinocerebellar ataxia; ^d^ asterisk signifies a nonsense mutation; ^e^ BTOP: bilateral temporooccipital polymicrogyria; ^f^ CMT4J: Charcot–Marie–Tooth disease type 4J; ^g^ YVS: Yunis–Varon syndrome; ^h^ IBMPFD1: inclusion body myopathy, Paget’s disease and frontotemporal dementia, type 1; ^i^ CMT2Y: Charcot–Marie–Tooth disease type 2Y; ^j^ IBMPFD3: inclusion body myopathy, Paget’s disease and frontotemporal dementia, type 3; ^k^ HSAN1A: hereditary sensory neuropathy type 1A; ^l^ SMAJ: spinal muscular atrophy, Jokela type; ^m^ IMMD: isolated mitochondrial myopathy, autosomal dominant.

## Data Availability

The data presented in this study are available in this article or in the Appendix A.

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
