# Peer review of "Yeast Models of Amyotrophic Lateral Sclerosis Type 8 Mimic Phenotypes Seen in Mammalian Cells Expressing Mutant VAPBP56S"

_biomolecules, 2023, doi:10.3390/biom13071147_

Round 1

Reviewer 1 Report

The research attempts to develop a new cellular model to study ALS.

Fig. 4I,

The indication of the confidence of IC50 determination from the dose response curves (+/- SEM) is missing.

Line 480-481

"Growth of scs2Δ scs22Δ + VAPBWT or VAPBP56S resulted in an intermediate level of growth that was not significant compared to either WT or scs2Δ scs22Δ cells."

The sentence is unclear and should be revised.

Author Response

We thank the reviewer for this helpful feedback. Our responses to each comment are noted below ini blue.

The research attempts to develop a new cellular model to study ALS.

Fig. 4I,

The indication of the confidence of IC50 determination from the dose response curves (+/- SEM) is missing.

We thank the reviewer for pointing out this omission. We have now included error values for the IC50 measurements as requested.

Line 480-481

"Growth of scs2Δ scs22Δ + VAPBWT or VAPBP56S resulted in an intermediate level of growth that was not significant compared to either WT or scs2Δ scs22Δ cells."

The sentence is unclear and should be revised.

We agree that this statement was unclear, and it was no longer necessary based on new experiments. Thus, we have removed it from our revised manuscript.

Reviewer 2 Report

In this manuscript, Stump and colleagues describe a novel yeast model of ALS8, characterized by mutations in the VAPB gene. The authors generated a double knockout of the yeast homologs Scs2 and Scs22 and showed significant defects to ER morphology and resilience to ER stress. Reintroduction of WT Scs2, but not ALS-like mutant Scs2 or VAPB, fully rescued those phenotypes. The authors thus propose this novel model as a useful tool to investigate ALS pathobiology and perform genetics or drug screenings to identify potential modifiers of the human disease. The data presented is solid and compelling, and the experiments are well controlled and sufficiently powered. My minor comments/concerns are listed below:

1. One of the morphological changes to ER that the author quantify is the size of inclusion-like accumulation of GFP-tagged Sec61. However, there is no data or discussion on the biological significance of such inclusions. The authors should further discuss or perform additional experiments to clarify the nature and relevance of the observed "inclusions". Alternatively, much less importance should be given to those inclusion in the interpretation of the data, in particular in relation to the effects of mutant VAPB.

2. All data on ER morphology collected are based on the expression of GFP-Sec61. A second marker should be used to validate  the morphological changes observed.

3. In both the "Results" section and "Discussion", the authors claim that the expression of WT human VAPB can fully compensate for the loss of endogenous Scs homologs in the double knock out yeast model (e.g. p.12, lines 455-456; p. 16, lines 590-591). However, Fig. 3C and Fig. 4G-H clearly shows that WT VAPB does not restore survival defects under ER stress conditions. The authors should revise the manuscript to make that clearer. 

3. The use of red or blue font in Table 1 is not explained in the table legend. 

Author Response

We thank the reviewer for their positive view of this work, and for the helpful criticisms provided. In the revised manuscript, we have attempted to address all points raised, as described below in blue text.

1. One of the morphological changes to ER that the author quantify is the size of inclusion-like accumulation of GFP-tagged Sec61. However, there is no data or discussion on the biological significance of such inclusions. The authors should further discuss or perform additional experiments to clarify the nature and relevance of the observed "inclusions". Alternatively, much less importance should be given to those inclusion in the interpretation of the data, in particular in relation to the effects of mutant VAPB.

We appreciate this thoughtful comment. While the monitoring editor indicated that we adequately described inclusions that have been previously documented in mammalian cells (lines 67-81), we agree that the nature and significance of related structures in our yeast model are not yet clear, even though they do recapitulate phenotypes seen in mammalian cells. We have not yet had the opportunity to perform ultrastructural analyses to determine the degree of similarity to ER inclusions in mammalian cell models, which will be included in future studies. Instead, in the revised manuscript we have made every effort to soften our language in referring to these as inclusion-like structures.

2. All data on ER morphology collected are based on the expression of GFP-Sec61. A second marker should be used to validate the morphological changes observed.

This suggestion is an important way to strengthen our findings. In the revised version of our study, we have added a new figure showing that two additional ER markers (Rtn1-mScarlet1 and DsRed-HDEL) co-localize with Sec61-GFP in all of the strains we used, and that all three markers are found in the inclusion-like structures seen in scs2∆ scs22∆, scs2∆ scs22∆ + scs2P51S, P58S, and scs2∆ scs22∆ + VAPBP56S cells (Results section 3.4 and newly-added Figure 4).

3. In both the "Results" section and "Discussion", the authors claim that the expression of WT human VAPB can fully compensate for the loss of endogenous Scs homologs in the double knock out yeast model (e.g. p.12, lines 455-456; p. 16, lines 590-591). However, Fig. 3C and Fig. 4G-H clearly shows that WT VAPB does not restore survival defects under ER stress conditions. The authors should revise the manuscript to make that clearer. 

We thank the reviewer for pointing out these inconsistencies. In the revised manuscript, we have moderated these statements and have made every attempt to clarify throughout.

3. The use of red or blue font in Table 1 is not explained in the table legend. 

We have made sure to include an explanation for the red and blue font (essential genes and essential gene combinations, respectively) in the legend for Table 2 (line 677) as well as in the discussion (lines 829-30).

Reviewer 3 Report

This represents an interesting study outlining the potential for the use of yeast models in ALS research. The particular mutant phenotype being investigated was that of ALS8 or the VAPB P56S mutation. An interesting ending to the paper outlines other possible ALS genes that could be investigated using the same model organism. The manuscript reads well and the work done is thorough. I have no qualms about the accuracy of the work but it would be of interest to record the confirmation of the sequences of the plasmids that were used in generating the models simply to ensure that the effects observed were entirely reliant upon the expected modifications. It remains of interest that significant growth changes were observed in the yeast along with other expected phenotypic changes and yet ALS is a late onset disorder. Can the authors shed any light upon this? 

It is obvious that the VAPB mutation can have possible impacts upon a number of metabolic pathways, (l81 onwards), and the model that has been developed, being single celled, may enable the complex interaction to be unravelled. 

I am intrigued that the presence of the mutant VAPB in the double knockout yeast seems to have some effect in ameliorating the phenotype. The normalized kinetic growth assays show some return to the WT level. This does not reduce significance but is an interesting change in metabolic behaviour. 

Overall the manuscript clearly demonstrates the potential for this model in ALS research. The work is clearly and accurately described but I would just like to confirm that all the introduced changes are as expected.

Author Response

We thank the reviewer for their positive assessment of our manuscript. In our revision, we have attempted to address all of the points raised (see blue text below). We are also intrigued by our observation of these phenotypes in an organism such as yeast on these timescales, given that most cases of ALS in humans develop later in life, including for ALS8. Although we can only speculate on this point at the present time, it is possible that milder stresses than are used in an experimental setting lead to progressive neuronal loss over time, possibly due to additive effects. We described this possibility in the revised discussion (lines 805-817).

It is obvious that the VAPB mutation can have possible impacts upon a number of metabolic pathways, (l81 onwards), and the model that has been developed, being single celled, may enable the complex interaction to be unravelled. 

We are particularly excited about this possibility – the genetic tractability of yeast combined with its short generation time make it particularly attractive as a model for understanding the underlying cellular basis of ALS8, and our observation of phenotypic similarities imply that highly conserved functions may be fundamentally important.

I am intrigued that the presence of the mutant VAPB in the double knockout yeast seems to have some effect in ameliorating the phenotype. The normalized kinetic growth assays show some return to the WT level. This does not reduce significance but is an interesting change in metabolic behaviour. 

This is an excellent point, and we are also intrigued about this observation. While we repeated the kinetic growth assays based on the comment below, we still observed the same trends for mutant VAPB in our models. We are not sure why this partial effect occurs, but we speculate that differences in interaction of human VAPB (versus yeast Scs2) with yeast homologs of its binding partners might be a factor. While VAPB clearly complements ER morphology defects seen in SCS-deficient yeast (e.g., Fig. 3A-B), the human protein may not interact as efficiently with proteins involved in ER stress responses. We provide speculation on this point in lines 738-741 of the revised manuscript.

Overall the manuscript clearly demonstrates the potential for this model in ALS research. The work is clearly and accurately described but I would just like to confirm that all the introduced changes are as expected.

We very much appreciate this comment, as it prompted us to repeat sequencing that was done in our original construction and validation of the Scs2 plasmids before the lab moved to our current institution – we did not have a copy of those original results on file and thus felt it prudent to repeat them out of an abundance of caution. We were surprised to discover a miscalled base that went undetected in our original sequence validation (a T-to-C transition at nucleotide 704 of the coding sequence, right at the boundary of a sequence comprising a set of C and T nucleotides [original sequence is CCTT, while the substitution resulted in sequence CCCT]). This would result in a missense L235P mutation; thus, during the revision period we completely remade the scs2P51S, P58S plasmids and all strains derived from them, fully validated the sequences, and repeated all experiments involving their use to verify that our observed results were indeed due to the P51S P58S mutations.

In the revised manuscript, we have replaced all data from Figures 2 and 4 with new experiments using the reconstructed strains. Since trials for the growth curve assays in Fig. 4 were paired to control strains (e.g., Figs. 4A and B), we completely replaced all data in this figure with new, paired experiments. While some values and statistical significances changed slightly and have been noted in the revised manuscript, the general conclusions based on the new experiments are in line with those of the initial submission. Thus, the phenotypes reported are indeed due to the P51S, P58S mutations.

With our revision, we have included a statement that plasmids were confirmed by Sanger sequencing. We also have added an appendix (Appendix A) listing the coding sequences used and indicating the altered sequences in mutant alleles.